Cognition 210 (2021) 104620

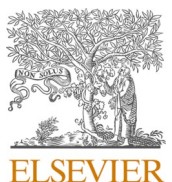

Contents lists available at ScienceDirect

# Cognition

journal homepage: www.elsevier.com/locate/cognit

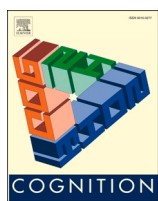

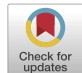

# What makes a language easy to learn? A preregistered study on how systematic structure and community size affect language learnability

Limor Raviv [a,b,*], Marianne de Heer Kloots [c], Antje Meyer [b,d]

[a] *Vrije Universiteit Brussels, Belgium*
[b] *Max Planck Institute for Psycholinguistics, the Netherlands*
[c] *University of Amsterdam, the Netherlands*
[d] *Radboud University Nijmegen, the Netherlands*

## ABSTRACT

Cross-linguistic differences in morphological complexity could have important consequences for language learning. Specifically, it is often assumed that languages with more regular, compositional, and transparent grammars are easier to learn by both children and adults. Moreover, it has been shown that such grammars are more likely to evolve in bigger communities. Together, this suggests that some languages are acquired faster than others, and that this advantage can be traced back to community size and to the degree of systematicity in the language. However, the causal relationship between systematic linguistic structure and language learnability has not been formally tested, despite its potential importance for theories on language evolution, second language learning, and the origin of linguistic diversity. In this pre-registered study, we experimentally tested the effects of community size and systematic structure on adult language learning. We compared the acquisition of different yet comparable artificial languages that were created by big or small groups in a previous communication experiment, which varied in their degree of systematic linguistic structure. We asked (a) whether more structured languages were easier to learn; and (b) whether languages created by the bigger groups were easier to learn. We found that highly systematic languages were learned faster and more accurately by adults, but that the relationship between language learnability and linguistic structure was typically non-linear: high systematicity was advantageous for learning, but learners did not benefit from partly or semi-structured languages. Community size did not affect learnability: languages that evolved in big and small groups were equally learnable, and there was no additional advantage for languages created by bigger groups beyond their degree of systematic structure. Furthermore, our results suggested that predictability is an important advantage of systematic structure: participants who learned more structured languages were better at generalizing these languages to new, unfamiliar meanings, and different participants who learned the same more structured languages were more likely to produce similar labels. That is, systematic structure may allow speakers to converge effortlessly, such that strangers can immediately understand each other.

## 1. Introduction

Languages differ greatly in how they map different meanings into morpho-syntactic structures (Dryer and Haspelmath, 2013; Evans and Levinson, 2009). Some languages appear to be relatively simple in terms of their morphology, while other languages are viewed as highly complex. For example, English makes minimal use of verb inflection to express grammatical relations: most English verbs have only one basic inflection paradigm to express time, such as adding [−ed] to express past tense, and this inflection is consistent across grammatical persons (i.e., *she* and *they* receive the same inflected form). Even verbs that are considered irregular in English (e.g., *sing, ring, buy, seek*) often follow a systematic inflectional rule (i.e., *sang, rang, bought, sought*). In contrast, Georgian has an elaborate set of verb inflection paradigms based on time, grammatical person, grammatical case, mood, and more (Hewitt, 1995; Imedadze and Tuite, 1992). Verbs in Georgian can take an astonishing number of different forms (estimated at around 200), and many verbs are truly irregular and follow unique rules, requiring speakers to learn the inflections of these verbs independently. Beyond such anecdotal examples, cross-linguistic studies have confirmed that languages differ in their degree of morphological complexity (Ackerman & Malouf, 2013; Bentz and Berdicevskis, 2016; Hengeveld and Leufkens, 2018; Lewis and Frank, 2016; Lupyan and Dale, 2010; McCauley & Christiansen, 2019).

This cross-linguistic difference in morphological complexity may have important consequences for learning: some languages may be easier to learn than others. This idea goes against a wide-spread axiom in the field of linguistics, which is that all languages are equally difficult to learn and take the same effort to acquire (Sweet, 1899). Recent work has challenged this axiom, and provided initial support for the premise that languages differ in their degree of learnability. Even though all languages are fully learnable in the long run (i.e., all children eventually

---

* Corresponding author at: Vrije Universiteit Brussels, Belgium.
  *E-mail address:* limor.raviv@mail.huji.ac.il (L. Raviv).

https://doi.org/10.1016/j.cognition.2021.104620
Received 1 March 2020; Received in revised form 14 January 2021; Accepted 27 January 2021
Available online 8 February 2021

acquire their native language, regardless of how complex it is), languages may still differ in how well they are learned in a fixed period of time, and/or in how long it takes to fully acquire them. For instance, corpus studies report that the trajectory of children's first language acquisition (L1) can vary across languages, such that the relative speed of learning differs for children acquiring different languages (Armon-Lotem et al., 2016; Bleses et al., 2008; Bleses et al., 2011; Dressler, 2003; Xanthos et al., 2011). Similarly, work on second language learning (L2) has shown that adults are better at learning some languages than others, suggesting that not all languages are equally learnable for adults given limited exposure (Kempe & Brooks, 2008; Kempe & MacWhinney, 1998). These differences in learning outcomes, learning trajectories, and proficiency have been related to various factors, including morphological complexity, i.e., the degree to which inflectional morphemes are informative, productive, and clearly marked. Specifically, languages with more regular, compositional, and transparent structures are generally considered to be easier to learn by both children and adults when compared to languages with opaque and irregular structures (DeKeyser, 2005; Dressler, 2003, 2010; Hengeveld and Leufkens, 2018; Slobin, 1985).

While there is no widely accepted way to measure morphological complexity, various metrics have been used – from counting the number of inflected word forms per lemma (Xanthos et al., 2011), to conditional entropy (Ackerman & Malouf, 2013; Winters et al., 2015), type/token ratio (McCauley & Christiansen, 2019), and the degree of regularity in the mapping between forms and meaning (Cornish et al., 2009; Tamariz, Brown, & Murray, 2010; Tamariz and Smith, 2008). Although the quantitative definitions of morphological complexity vary across researchers, its descriptive notion is relatively stable. Generally speaking, a language is considered to be simpler if it is compositional, regular, and transparent, i.e., if there are systematic one-to-one relations between units of meanings and units of form (DeKeyser, 2005; Hay and Baayen, 2005; Hengeveld and Leufkens, 2018). For example, the word [*walked*] consists of two parts: the verbal stem [*walk*] and the past tense morpheme [*-ed*], which are combined in a transparent way to express the act of walking in the past. In comparison, the irregular past form [*bought*] cannot be as easily divided into separate bits, making it more holistic and opaque. Similarly, a language is considered to be more complex if the meanings of words are not directly predictable from their constituents. Such opacity can stem from multiple sources, such as having redundant or optional marking, syncretism, and/or a high prevalence of inconsistencies and irregularities. In this sense, more complexity is seen as the result of having less transparency. Complexity can also stem from having multiple inflectional paradigms and many obligatory grammatical rules. As such, the relation between complexity and transparency is not always straight-forward (Kempe and Brooks, 2018; Kempe & MacWhinney, 1998). For example, Russian has complex and elaborate inflectional paradigms with multiple grammatical cases, which are nevertheless transparent and informative; in contrast, German has considerably simpler paradigms with fewer grammatical cases, but high levels of syncretism that render the system fairly opaque and uninformative. In any case, the main theoretical notion of linguistic complexity incorporates the idea that more regularity, more transparency, and more compositionality are simpler and should therefore be beneficial for learning.

Intuitively, it seems reasonable that languages with more regular and compositional morphology will be easier to learn, given that they generally allow learners to derive a set of productive rules rather than memorizing individual forms (Kirby, 2002; Zuidema, 2003). This intuition is computationally and mathematically supported by information theory and set redundancy compression models, which show that data with systematically recurring elements can be more efficiently compressed into fewer bits (Karadimitriou, 1996; Kortman, 1967). Moreover, the postulated positive relation between systematicity and learnability is based on general characteristics of the learning system, such as a domain-general cognitive bias in favor of simplicity (Chater

and Vitányi, 2003; Culbertson and Kirby, 2016). As such, the learning advantage of systematicity is expected to hold across one's lifespan and be present in learners of all ages. In other words, both children and adults should benefit from having more regularity in their input. We return to this point in the Discussion.

However, the causal relationship between linguistic structure and language learnability in both children and adults is currently untested. Very few studies have attempted to examine this link by investigating learning difficulty as a function of linguistic complexity. Only a handful of correlational and experimental studies have examined learning outcomes and learning trajectories in natural languages that differ in their morphological complexity. These studies exhibit a mixed patterns of results: some report slower acquisition and worse overall proficiency for natural languages that are more morphologically opaque (Kempe & Brooks, 2008; Kempe & MacWhinney, 1998; Slobin, 1985), while others report similar learning rates across different natural languages (Armon-Lotem et al., 2016; Braginsky et al., 2019), or the opposite pattern altogether, i.e., that morphologically complex languages are acquired faster (Dressler, 2003; Xanthos et al., 2011). There are various potential reasons for these conflicting findings, such as different complexity metrics used across studies, different variables of interest, different age groups of learners, etc. In any case, this mixed pattern of results highlights the lack of decisive empirical evidence and consensus on the relation between systematicity and learnability in natural language.

No study to date has systematically compared the acquisition of a broad yet comparable range of morphological structures using a controlled experimental paradigm such as artificial language learning. As such, there is no direct empirical evidence that languages with more regular and transparent structures are indeed easier to learn by children or adults. While direct empirical evidence for this argument is lacking, two studies provide initial support for this assumption. Brooks et al. (1993) and Monaghan et al. (2011) both conducted artificial language learning experiments to test the acquisition of languages that differed in their degree of sound-systematicity, i.e., the mapping between forms and categories. In these studies, participants were trained on a miniature vocabulary containing two word classes, corresponding to grammatical gender (Brooks et al., 1993) or to actions/objects (Monaghan et al., 2011). In the arbitrary condition, there were no similarities between the words' phonological forms and their grammatical classes, such that sounds were distributed randomly between the two classes. In Monaghan et al. (2011), who tested adults, this condition was contrasted with a fully systematic condition, in which words from different grammatical classes contained distinct sounds (e.g., words for objects contained fricatives while words for actions contained plosives) and with a partially systematic condition in which members of each noun class shared a subset of phonological features. In Brooks et al. (1993), who tested both children and adults, the arbitrary condition was only contrasted with a partially systematic condition. In both studies, participants were significantly better at learning the distinction between the two categories when there was full or partial systematicity in the mapping between forms and meanings, i.e., when there was a phonological cue indicating the nouns' grammatical category. Brooks et al. (1993) showed that children and adults were similarly affected by the degree of systematic mapping between phonological forms and grammatical categories, and benefitted from having systematic cues in their input. These findings provide initial support for the idea that learning outcomes can be affected by the degree of systematic structure (at least in terms of mapping sounds to grammatical classes). But since these studies did not directly test the effect of morphological complexity or compositionality, they are not sufficient for concluding that compositional, transparent, and regular languages are indeed easier to learn.

The current study aimed to fill this gap and experimentally test the learnability of artificial languages that vary in the degree of systematic structure (i.e., in how transparent, compositional, and predictable the mapping between meanings and labels is). We examined adults' acquisition of miniature languages that had been created by big and small

groups of interacting participants in a previous group communication experiment (Raviv et al., 2019b). Specifically, adult learners were trained on input languages that varied in their degree of systematic structure (ranging from completely unstructured languages to highly systematic languages) and in their group size origin (whether they had been created by big or small groups). After training, participants were tested on their knowledge of the input language in a memory test (measuring their reproduction accuracy on learned items) and in a generalization test (measuring their ability to label new, unseen items). We compared adults' learning of the input languages with two questions in mind: (1) are more systematic languages easier to learn? And (2) are languages created by bigger groups easier to learn, above and beyond the effect of systematic structure?

Answering question (1) was the main goal of the current study. This research question was motivated by the literature discussed above, as well as by its importance to influential theories on language evolution and language diversity, which are discussed below. To preview, with respect to Question (1), our findings confirmed the link between systematicity and learnability, showing that regardless of group size origin, the more structured languages were learned significantly better and faster by the adult learners. Moreover, we found that systematic structure was advantageous not only for learning, but also for generalization productivity. Question (2) was motivated by studies suggesting that visual signals (i.e., drawings) created by bigger groups are processed, learned, and reproduced faster by new individuals compared to signals created by pairs, despite these signals being equally complex (Fay et al., 2008; Fay and Ellison, 2013). Therefore, we considered the possibility that, even when equating for the degree of structure in the language, languages that evolved in bigger groups may be easier to learn due to other features. This hypothesis by no means entails that languages of bigger communities are *better* than those of smaller communities: all natural languages are equally good at expressing messages, and being "*easy to learn*" does not entail any quality judgment (for more on this point, see Gil, 2001; Raviv, 2020, pp. 223–224). To preview our results, with respect to Question (2) we found no evidence of additional benefits for languages created by bigger groups beyond their degree of systematic structure.

The hypothesis that transparent and regular grammars are more easily learned, at least by adults, is essential for the theoretical reasoning in at least two fields: (a) language evolution simulations on the emergence of linguistic structure during cross-generational transmission, and (b) the social origin of cross-linguistic diversity. In both fields, the postulated causal relationship between systematic linguistic structure and language learning serves as a crucial assumption that underlies much of their motivations and conclusions. As such, it is important to validate this link.

In the first line of research, language evolution models explicitly argue that compositional structure emerges as a consequence of learnability pressures combined with expressivity pressures, and that compositional structure facilitates the accurate transmission of languages over multiple generations of learners, who would struggle to learn a holistic and unstructured lexicon (e.g., Cornish et al., 2009; Kirby, Cornish, & Smith, 2008; Kirby et al., 2015; Smith, 2011). At the core of this literature is the assumption that systematic structure is advantageous for learning in general – be it for a human child, an adult, a primate, or a completely simulated computer agent – and thus promotes the emergence and accumulation of structure over generations of learners. Using iterated learning and diffusion chain paradigms, multiple studies have reported that artificial languages become more compositional (as reflected by more systematic form-meaning mapping) and more faithfully reproduced (as reflected by fewer transmission errors) over generations (Beckner et al., 2017; Kirby, Cornish, & Smith, 2008; Raviv and Arnon, 2018). In this field, the emergence of compositional languages is directly attributed to learning pressures: more systematic and predictable signals are presumably favored over generations *because* they are learned better (i.e., there are fewer unique forms

to remember, making languages easier to reproduce (Cornish, 2010; Tamariz and Kirby, 2016). Moreover, compositional languages are argued to be advantageous for generalizations, allowing learners to overcome the "poverty of stimulus" (Kirby, 2002; Kirby, Smith, & Brighton, 2004; Zuidema, 2003): since learners must acquire their linguistic competence from finite and partial input, languages with more regular and transparent structures should be favored because they allow learners to easily refer to new, unfamiliar meanings using the same system. In other words, iterated learning studies assume a close and causal relationship between linguistic structure and learnability, and the hypothesized mechanism behind the emergence of structure strongly relies on the intuition that more systematic languages are easier to learn and are more generalizable.

Accordingly, iterated language learning studies typically report a simultaneous increase in both systematic structure and learnability over generations of learners, which is taken as evidence that more structured languages are easier to learn (e.g., Beckner et al., 2017; Kirby, Cornish, & Smith, 2008). Yet ttthese studies typically do not examine this relation directly, and rely on correlational evidence. As such, iterated language learning paradigms have not directly confirmed the *causal* role they assume between linguistic structure and learning, beyond the mediating variable of generation number. Nevertheless, there is some evidence in support of the correlation between accuracy and systematicity in such paradigms. For example, Tamariz and Smith (2008) found that participants who learned languages with more regular form-to-meaning mappings also produced languages with more regular form-to-meaning mappings, but participants' accuracy in learning the input language was not reported. Two other studies reported a significant correlation between learning accuracy and producing systematic structure, albeit in the opposite postulated direction of causality: transmission error was a significant negative predictor of linguistic structure across all generations of learners, such that participants who showed better learning of the input language also introduced more linguistic structure when reproducing the language (Johnson et al., 2020; Raviv and Arnon, 2018). Finally, the results of one iterated learning study suggested that linguistic structure and learnability are not always related: Berdicevskis (2012) found that even though artificial languages became more compositional over generations of learners, they did not become more learnable: there was no significant increase in reproduction fidelity over generations despite the increase in systematic structure, and there was no correlation between how compositional languages were and how accurately they were learned. That is, the increase in linguistic structure did not facilitate learning. Together, these findings strengthen the need for conducting a careful examination of the causal relation between language learnability and systematicity.

As for the second line of research, on the social origin of linguistic diversity, cross-linguistic work has found that languages spoken by big communities are typically less morphologically complex than languages spoken by small communities – a finding that has been typically attributed to learnability pressures caused by the presence of more adult second-language learners in larger communities (Bentz et al., 2015; Bentz and Winter, 2013; Lupyan and Dale, 2010). Specifically, the inverse correlation between morphological complexity and population size has been argued to be driven by the higher proportion of non-native adult speakers in larger communities, and the difficulty of adult L2 learners in acquiring morphologically complex and opaque languages (Bentz and Berdicevskis, 2016; Dale & Lupyan, 2012; Lupyan and Dale, 2010, 2016; McWhorter, 2007; Trudgill, 1992, 2002, 2009). In other words, the reduced morphological complexity observed in the languages of larger communities is hypothesized to be the direct result of the postulated relationship between linguistic structure and learnability: because adult second language learners typically struggle with learning languages with complex structure, languages adapt and simplify in the presence of many such learners (which is more typical of bigger communities).

This line of reasoning includes two assumptions: (a) that the

presence of adult non-native speakers leads to morphological simplification, and (b) that morphologically simpler languages are in turn advantageous for learning and use by adults. The first assumption, i.e., that the presence of adult non-native speakers leads to morphological simplification, has received consistent support from multiple sources. For example, artificial language learning studies report that adults' non-native language learning (as simulated by insufficient exposure during learning) leads them to simplify morphologically complex languages (Atkinson et al., 2018b; Bentz and Berdicevskis, 2016), and that such simplifications may increment during the process of cross-generational transmission (Berdicevskis & Semenuks, 2020). In addition, work on koineization (dialect mixing) and language contact has shown that languages tend to simplify when several dialects come together, or due to long-term accommodation to second language learners as in migration scenarios (Kerswill & Williams, 2000; Trudgill, 1992, 2002, 2008, 2009). Supporting the latter point, experimental work confirms that native speakers tend to adapt more to the syntactic choices of non-native than native confederates, even when they produce simpler, ungrammatical sentences (Loy and Smith, 2019). Thus, there is ample empirical evidence for the assumption that the presence of adult non-native speakers can induce language simplification.[1] The second assumption (i.e., that morphologically simpler languages are advantageous for learning and use by adults), has not been tested directly. Nevertheless, it receives indirect support from the literature on second language learning, which demonstrates that adults generally struggle with learning and using morphology in a second language: adults L2 speakers typically show optional or variable use of verbal and nominal inflections related to case marking, tense, agreement, aspect, and gender marking (DeKeyser, 2005; Haznedar, 2006; Parodi, Schwartz, & Clahsen, 2004), and learn faster when languages exhibit more reliable morphological cues (Kempe & MacWhinney, 1998).

An alternative explanation for the documented correlation between morphological complexity and community size is that, instead of being mediated by the proportion of adult non-native speakers and their difficulty in language learning, it is directly derived from differences in community size (Nettle, 2012; Raviv et al., 2019b; Wray and Grace, 2007). According to this hypothesis, the total number of speakers in the community can affect language structure during peer-to-peer diffusion, and there is no need to assume the prevalence of second language learning as a mediating factor: big communities might favor simpler and more transparent linguistic structures simply because they are big. The general idea is that members of bigger communities generally face a greater communicative challenge as they need to interact with more people with whom they have less shared history. Consequently, members of bigger communities may be under a stronger pressure to develop more systematic and transparent languages that are presumably easier to remember, and that can in turn facilitate convergence between strangers (Wray and Grace, 2007). This hypothesis is supported by typological studies on newly emerging sign languages, which show that within the same time period, languages that developed in bigger and sparser communities were more systematic and more conventionalized than languages developed in smaller and tightly knit communities (Meir, Israel, Sandler, Padden, & Aronoff, 2012; Meir and Sandler, 2019). Furthermore, this hypothesis was recently tested experimentally using a group communication paradigm, in which groups of four or eight interacting participants needed to create new artificial languages to communicate with each other about different novel scenes (Raviv et al., 2019b). Results showed that larger groups developed more systematic and compositionally structured languages over the course of the experiment, and did so faster and more consistently than small groups.

Furthermore, the emergence of more systematic and compositional languages in larger groups was advantageous for communication, with more linguistic structure being associated with better convergence In this study, community size alone had a causal role in shaping the emergence of more systematic linguistic structure, and its results suggest that more systematic languages may facilitate convergence between more individuals.

Notably, this line of reasoning still assumes that systematicity is advantageous for adults: systematic languages are presumed to be selected for in bigger groups *because* they are more efficient for learning and use. But is this indeed the case? Would the languages developed by larger groups in Raviv et al. (2019b) be more easily acquired by naïve individuals? This question draws a direct link between the two literatures discussed above, i.e., iterated language learning and the social origin of linguistic diversity: if larger communities tend to have more systematic languages (as was also shown in the research on linguistic diversity; e.g., Lupyan and Dale, 2010), and if more systematic languages facilitate language learning by the next generation of learners (as argued by iterated language learning theories; e.g., Kirby, Cornish, & Smith, 2008), then the languages of larger communities should be more easily acquired by naïve adults.

We also set out to investigate whether there may be additional advantages to languages developed in bigger groups, above and beyond their degree of systematicity. This question was motivated by two sets of findings. First, computational models of iterated learning have shown that languages adapt to fit agents' cognitive biases over generations, such that agents' weak individual tendencies become amplified as languages are transmitted by more and more individuals (Kirby, Smith, & Brighton, 2004, Kirby, Dowman, & Griffiths, 2007; Reali and Griffiths, 2009; Smith, 2011). This result suggests that languages that evolved in larger populations may be even more fitted to individuals' cognitive and learning biases, as they have passed the processing filter of more individuals and have been used by more people. As such, it is possible that languages developed in larger groups would have additional properties (e.g., word form, affect) that would facilitate their learning even when having similar degrees of systematic structure as languages of small groups. Second, earlier studies suggest that this is the case for non-linguistic visual signals, namely drawings. When groups of eight people played multiple rounds of Pictionary, their final drawings were superior to those developed by pairs in terms of their learnability and processing by new individuals, despite being comparable in visual complexity (Fay et al., 2008; Fay and Ellison, 2013): adult learners were more accurate in guessing the meanings of drawings that evolved in larger groups, and were able to learn them faster, recognize them faster, recall them faster, and reproduce them with better fidelity compared to drawings that evolved in pairs. This advantage was attributed to the large groups' drawings being more iconic, i.e., having more transparent form-to-meaning mappings. Fay et al. (2008) concluded that the greater "fitness" of signs developed by big communities was derived from the increased diversity of potential signs: larger groups have a greater pool of variants to draw from, allowing for the selection of simpler signs. If such reasoning extends to language, then the greater input variability in the big groups in Raviv et al. (2019b) may similarly benefit learners in the long run by favoring the selection of more transparent signs. Although there is no direct evidence that languages of larger communities are easier to learn, one study tested the effect of group size on the complexity and transparency of linguistic conventions that were created by two vs. three individuals (Atkinson et al., 2018a). In that study, dyads and triads used English to describe novel icons to each other, and their final descriptions were transmitted to naïve learners who had to match them to their referents. Atkinson et al. (2018) found that matching accuracy did not differ significantly between the two- and three-person conditions, providing no evidence that larger communities create more transparent descriptions. However, it is possible that the group size manipulation used in that study (i.e., comparing two to three people) was not sufficiently strong, and/or that examining descriptions in

---

[1] This is, of course, not the only source of language change in the real world. There are many other paths and motivations for language change that do not involve second language learners (e.g., random drift, prestige bias, reanalysis, grammaticalization, etc.).

participants' pre-established language does not give rise to transparency differences. Therefore, it is still possible that novel communication systems developed in big groups are easier to learn after all. Again, this would not imply that languages of bigger communities are in any sense better than those of small communities.

## 2. The current study

The goal of the current study was to experimentally test the causal relationship between language complexity and language learnability, as well the role of community size in shaping such patterns. To this end, we used an artificial language learning paradigm in which adults needed to learn a new miniature language with labels for describing different types of novel scenes (see Procedure). After training, participants were tested on their knowledge of the input language in two ways: (a) a memory test, testing participants' reproduction accuracy on the scene-label pairings; and (b) a generalization test, testing participants' ability to label new, unseen scenes.

Participants were trained on different input languages, all of which had been created in a previous experiment by real groups of either four or eight interacting participants playing a communication game (Raviv et al., 2019b). We contrasted learning across several conditions by selecting ten different input languages, which varied in their degree of systematic structure and in their group size origin in a counter-balanced design, while being relatively similar in their average word length and internal confusability (see Materials). For example, a participant could learn a high-structured language, a medium-structured language, or a low-structured language, either of which could have been created by a big group or a small group.

In order to promote open-science and increase the transparency and credibility of our research, the entire study (e.g., design, procedure, predictions, analyses plans, etc.) was pre-registered on OSF and is available online: https://osf.io/9vw86/. Additionally, all the data collected in this experiment and the scripts for generating all analyses can be openly found at https://osf.io/d5ty7/.

For our confirmatory analyses, the main prediction was that linguistic structure would significantly affect language learnability, such that more compositional languages with systematic form-to-meaning mappings would be easier to learn (i.e., more accurately learned). Therefore, we expected that participants who learned more structured languages would show higher reproduction accuracy. We also hypothesized that group size would have an additional effect on language learnability, beyond the effect of linguistic structure: languages created by bigger groups were postulated to be easier to learn compared to languages created by small groups, above and beyond their degree of systematic structure. Therefore, we expected that across all structure levels, participants who learned languages that were created by big groups would show higher reproduction accuracy. We also planned to carry out exploratory analyses to examine the speed of learning across conditions, and to test the effect of linguistic structure and community size on participants' ability to generalize the language to a new set of meanings.

## 3. Methods

### 3.1. Participants

We analyzed data from 100 adults (79 women) between the ages of 18 and 35 (mean age = 22.9y). This sample size was determined in advance using a power analysis based on pilot data and power simulations for a range of possible effect sizes (see Appendix A). We tested two additional participants who did not complete the experiment, and so their data was not included in the analyses. Each participant was paid 10€. All participants were native Dutch speakers, and had no reported visual or reading difficulties. Ethical approval was granted by the Faculty of Social Sciences of Radboud University Nijmegen.

### 3.2. Materials

We selected ten languages from a bigger database of artificial languages, which were created in a previous experiment (Raviv et al., 2019b). The full database contained 144 languages that were created by individual participants in either small or larger groups after completing a group communication game. Each language consisted of 23 scene-label pairings. i.e., 23 written labels that corresponded to 23 dynamic visual scenes. The scenes varied along three semantic dimensions: shape, angle of motion, and fill pattern. Each scene consisted of one out of four possible shapes, moving repeatedly in a straight line from the center of the screen in a given direction. Additionally, each scene had a unique blue-hued fill pattern. There were three versions of the stimuli, which differed in the distribution of shapes and their associated angles.

Each language in the database had a *structure score*, which reflected the degree of systematic mapping between labels and meanings in the language (Kirby, Cornish, & Smith, 2008; Kirby et al., 2015; Raviv et al., 2019a). The structure score for each language was calculated as the correlation between the pair-wise semantic distances between scenes' features and the pair-wise string distances between their labels. First, we calculated the semantic differences between different scenes, resulting in a similarity matrix for all pairs of scenes in the language. This was done using Hamming distances, in the following way: First, two scenes had a semantic difference of 1 if they differed in shape, and a semantic difference of 0 if they included the same shape. Second, the difference between two scenes' angles was calculated and divided by the maximal distance between angles (180 degrees) to yield a continuous normalized score between 0 and 1. Then, the difference scores for shape and angle were added, yielding a possible semantic distance between 0.18 and 2 for each pair of scenes in the language. Next, we calculated the string differences between all pairs of labels in the languages using normalized Levenshtein distances, which is the minimum number of character changes (insertions, deletions or substitutions) needed in order to transform one label into the other, divided by the length of the longest label. This resulted in a similarity matrix for all pairs of labels in the language. Finally, the two sets of pair-wise distances (i.e., string distances and meaning distances) were correlated using the Pearson product-moment correlation, yielding a measure of systematic structure, i.e., whether similar meanings were expressed using similar strings.

This continuous measure was divided into five equally sized bins of possible structure scores[2]: low structure (0.0–0.2), low-medium structure (0.2–0.4), medium structure (0.4–0.6), medium-high structure (0.6–0.8), and high structure (0.8–1.0). Fig. 1 gives a general description of the structural properties of languages in each structural bin, along with an illustration. Low structure scores reflect the absence of systematic mapping between labels in the language and their corresponding scenes, resulting in a holistic lexicon where labels seem to be randomly assigned to the scenes regardless of their semantic features (see Fig. 1 for an illustration). In low structured languages, each scene has an opaque label that cannot be decomposed into small components based on scenes' shape or direction of motion. In contrast, high structure scores reflect the existence of systematic mappings between meanings and labels, resulting in compositional languages in which similar semantic features are expressed using similar part-words (see Fig. 1 for an illustration). Specifically, a highly systematic language would include a consistent part-word for describing each of the four shapes (e.g., "*tup*" for Shape 1 and "*fest*" for Shape 2), and a consistent part-word for describing the direction of motion (e.g., "*o*" for up, "*i*" for right, and "*oi*" for up-right). In addition to the structure score, we characterized

---

[2] Although correlations can potentially range from −1 to 1, there were no languages with a correlation below 0 (i.e., a languages with "anti-systematic" or "counter-systematic" mapping between labels and scenes). The structure scores of the languages in the data set ranged from 0.07 (i.e., an unstructured, holistic language) to 0.9 (i.e., a fully systematic, compositional language).

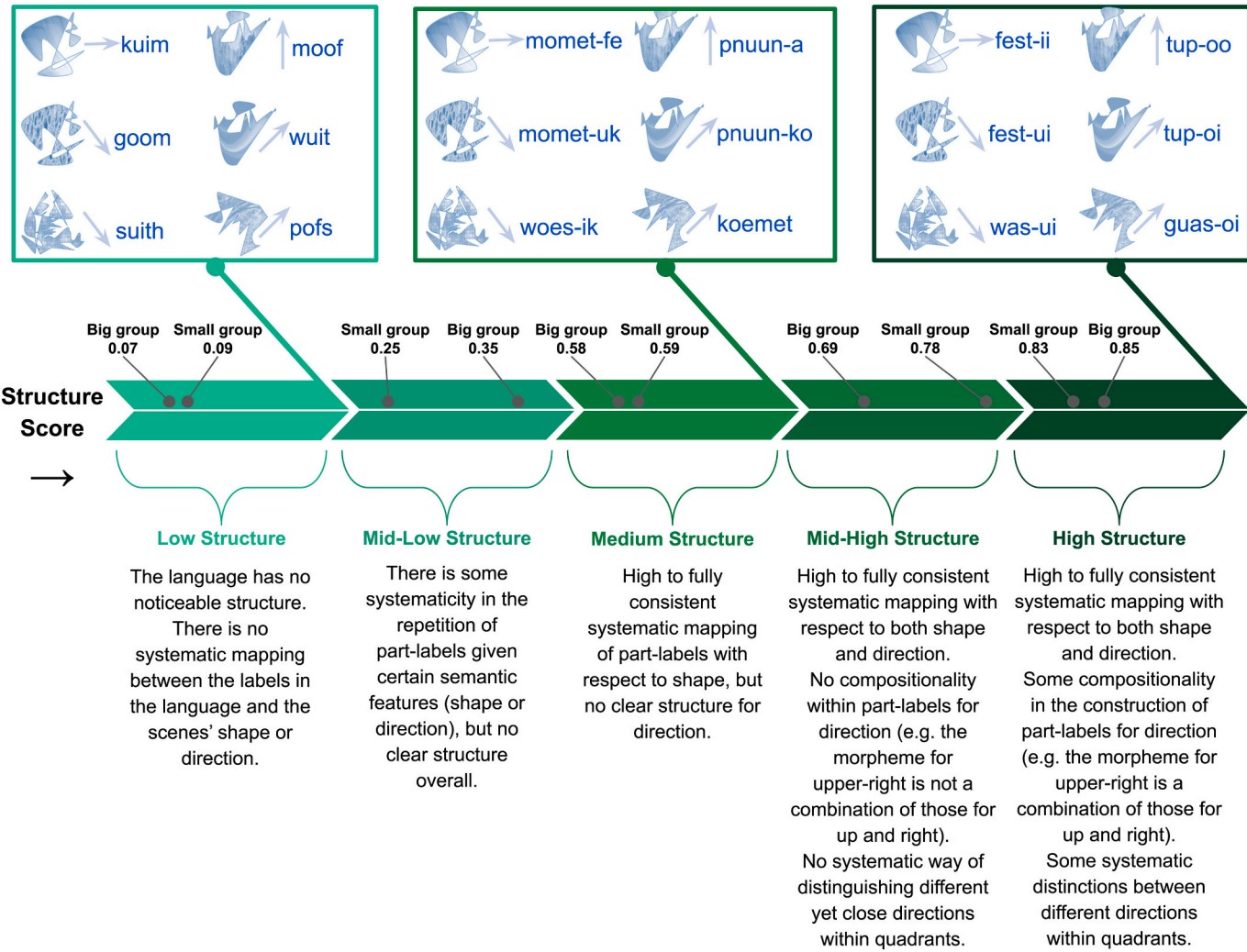

**Fig. 1.** An illustration of the structure levels of input languages learned by participants in the experiment. The axis represents languages' structure scores, ranging from 0 to 1. For descriptive purposes, this continuous measure can be divided into five equally sized bins: low structure (0.0-0.2), low-medium structure (0.2-0.4), medium structure (0.4-0.6), medium-high structure (0.6-0.8), and high structure (0.8-1.0). Each bin can be characterized by a different degree of systematicity, which is described verbally below it. We included illustrations of three miniature lexicons: a language with low structure in light green, a language with medium structure in green, and a language with high structure in dark green. For example, in the low structured language, there is no similarity between the labels for scenes with similar shapes (e.g., moof, wuit) or for scenes with similar directions (e.g., wuit, pofs). In the high structured language, part-labels are consistently associated with a given shape (e.g., fest, tup) or with a given direction (e.g., ui, oi). The direction morphemes are also compositional, and are comprised of two meaningful parts: for example, the morpheme for the direction up-right (oi) is a combination of the morpheme for up (o) and the morpheme for right (i). The grey dots on the axis point to the structure scores of ten specific languages originally created in a group communication game (Raviv et al., 2019b) which were selected as the input languages for this experiment. From each of structure bin, we selected one language that was created by a small group and one language that was created by a big group.

each language using two other measures: *average word length*, i.e., the average number of characters in the language's labels; and *confusability*, i.e., the average normalized Levenshtein distance between all possible pairs of labels in a language, capturing the phonological similarity across all labels in a given language.

We then selected ten languages from the database to be used as input languages for the current study (Fig. 1; see Appendix B for the full list of stimuli). Specifically, we picked two languages from each of the five structure score bins described above: one language that was created by a small group, and one language that was created by a big group. This resulted in a 2 × 5 factorial design, with the two factors being group size (with two levels: big vs. small) and structure score (with five levels of structure, ranging from low to high). Note that although we used these descriptive bins to select our input languages, structure score was treated as a continuous variable in all our analyses (ranging from 0.07 in the low structure bin, to 0.84 in the high structure bin). Ten participants

were assigned to learn each of the ten input languages using a pre-made randomization list.

Since we wanted to ensure that differences in language learnability can indeed be attributed to their structural properties and/or group size origin, we picked languages that were comparable in several ways. First, all languages fell within a reasonably similar range of average word length and confusability scores. Under the assumption that longer and more confusable words are harder to learn (Laufer, 2009; Willis and Ohashi, 2012), we chose languages from the lower half of the distributions of these two measures, i.e., languages with relatively short words (i.e., between 4 and 7 characters) and relatively low confusability (i.e., between 0.14 and 0.37). Second, languages in the same structure bin were comparable in terms of their descriptive grammatical properties and had similar types of consisted mappings (as judged by the authors; see Fig. 1 and Appendix B). Third, languages within the same structure bin had similar numbers of irregulars, as counted by the authors. Fourth,

across the different structure bins, differences in structure scores were balanced with group size, so that it was not the case that one group size condition was consistently higher/lower in structure compared to the other. The structure scores of the selected languages can be seen in Fig. 1. For the full set of input languages and their detailed descriptions, see Appendix B.

Finally, we created 13 new scenes for each stimuli version. These additional scenes were not included in the learning phase and the memory test, and were presented to participants for the first time during the generalization test. The new scenes varied along the same semantic dimensions as the 23 original scenes in the input languages, and were comprised of one of the four possible shapes moving in one of the possible directions. Each new scene included a new combination of shape and angle of motion (with each of the four shapes appearing in at least two new scenes), and a completely different blue-hued fill pattern. That is, the new scenes matched the general meaning space of the language, but included new combinations of shape, direction, and fill pattern which were unfamiliar to the participants and not present beforehand.

### 3.3. Procedure

Participants were told they were about to learn a new fantasy language ("Fantasietaal" in Dutch) to describe different scenes of moving shapes, and that their goal was to learn the language as best as they could in order to succeed in a subsequent test. The experiment consisted of two phases: (1) a learning phase, which was comprised of three leaning blocks with a similar procedure; And (2) a test phase, which was comprised of two parts, i.e., a memory test and a generalization test. Example screenshots of each phase of the experiment, along with detailed descriptions of the accompanying instructions and procedure, can be found at https://osf.io/mkv5r/.

The learning phase consisted of three blocks. The first learning block included half of the language (12 scene-label pairings), the second learning block included the other half of the language (11 scene-label pairings), and the third learning block included the entire language (23 scene-label pairings), effectively combining the first and second learning blocks. Each input language was randomly divided into two halves in advance, so that the set of target scenes in the first two blocks was identical for all participants in a given condition. We decided to divide the languages in half following the results of a pilot study, which suggested that participants can get overwhelmed and demotivated when exposed to all 23 scene-label pairings at once. The order of appearance of target scenes within a given block and during the test phase was randomized separately per participant at the beginning of the experiment.

Each learning block comprised of three tasks: passive exposure, guessing, and production. During passive exposure, participants were exposed to scene-label pairings one by one in a random order, with each target label appearing on the screen together with its corresponding scene for the duration of 10 s. In this task, participants only had to look at their screen and try to remember the scene-label pairings. In the guessing task, participants were presented with the target labels one by one in a random order, and needed to select the scene to which that label referred to from a set of possible scenes. In the first two blocks, this set included four scenes (i.e., the target scene and three distractors), while in the third block this set included eight scenes (i.e., the target scene and seven distractors). The distractors were randomly selected for each participant and for each trial from the set of possible scenes in that block. Participants received feedback after each guess indicating whether they were right or wrong, along with the target label, the correct scene, and the scene they selected in case it was different than the correct scene. In the production task, participants were presented with the target scenes without labels one by one in a random order, and needed to type their correct labels using their keyboard. Participants' letter inventory was restricted, and matched the letter inventory of the

original input languages from Raviv et al. (2019b): it included a hyphen, five vowel characters (a,e,i,o,u), and ten consonants characters (w,t,p,s, f,g,h,k,n,m), which participants could combine freely. Participants received feedback after each production, along with the target scene, the correct label, and the label they typed in case it was different than the correct label.

In the first two learning blocks, which included only half the language, each of the three tasks (i.e., passive exposure, guessing, production) was repeated twice with all the available target scenes-label pairings for that block, so that each scene-label pairing appeared twice in each task and six times in total. In the third learning block, which included the entire language, each task was repeated once, so that all scene-label pairings appeared once in each task and three times in total. This resulted in a total of nine exposures per scene-label pairing during the entire learning phase: three times during the passive exposure task, three times during the guessing task, and three times during the production task.

Following the learning phase, participants completed a test phase. The first part of the test phase was a memory test, in which participants demonstrated how well they had learned the input language. During the memory test, participants were presented with each of the 23 target scenes without labels one by one in a random order, and typed in a label for them. The second part of the test was a generalization test, in which participants were asked to use the language they had just learned to label new scenes that they had not seen before. Participants were presented with 13 unfamiliar scenes (see Materials) without labels one by one in a random order, and typed in a label for each of them based on their acquired knowledge of the Fantasy language. Participants were asked to label the new scenes as if they were communicating to another person, who had learned the same Fantasy language as they did but knew no other language (i.e., no use of Dutch, English, or any other language was allowed). No feedback was provided during the memory and generalization tests, and participants' letter inventory was restricted in the same manner as in the production phase.

After the test phase, participants filled out a questionnaire about their performance in the experiment, including questions such as "How hard was it to learn the Fantasy language?", and "Did you notice anything about the structure of the Fantasy language during the experiment?". Finally, all participants were debriefed by the experimenter.

## 4. Measures

### 4.1. Binary Accuracy

This measure reflects whether participants were correct or incorrect on a given trial during the learning phase or the memory test, and is calculated as binary response accuracy. If participants produced/ guessed the target label correctly, accuracy equaled 1; otherwise, it equals 0.[3]

### 4.2. Production similarity

This continuous measure reflects how closely participants reproduced their input language by measuring the similarity between a target label (i.e., an original label as it appeared in the input language) and the corresponding label produced by a participant in production trials (during the learning phase and during the memory test). For each production trial, we calculated the normalized Levenshtein distance between the label produced by the participant and the original input label. The normalized Levenshtein distance is the minimum number of

---

[3] In cases where the target label described more than one scene (i.e., homonym), participants' accuracy in guessing trials (during the learning phase) would equal 1 if they had guessed any one of the possible scenes associated with that target label.

character changes (insertions, deletions or substitutions) needed in order to transform one label into the other, divided by the length of the longest label. This distance was subtracted from 1 to represent string similarity, i.e., how much the labels participants produced resembled the labels they had learned. High production similarity indicates good reproduction fidelity, with participants producing labels that are similar to those they learned (i.e., a score of 1 indicates that the produced label matched the target label exactly). Low production similarity indicates poor reproduction fidelity, with participants producing labels that are very different from those they learned.

### 4.3. Guessing similarity

This continuous measure reflects how well participants learned the label-scene mapping in the input language by measuring the similarity between the target scene (i.e., the correct scene given a specific label) and the scene selected by the participant during guessing trials. We used Hamming distances to quantify the semantic differences between the selected scene and the target scene based on the differences in scenes' shapes and direction of motion. This measure was calculated in a similar way to the semantic distances used for calculating the structure score (see Materials). High guessing similarity indicate that, given a target label, participants guessed a scene which was similar to the target scene in terms of its features (i.e., a similarity score of 2 indicates that the selected scene matched the target scene exactly). Low guessing similarity indicates that, given a target label, the participant's guess was very different from the target scene (i.e., a similarity score of 0 indicates that the participant selected a maximally different scene with a different shape going to the opposite direction).

### 4.4. Generalization score

This continuous measure reflects the degree of similarity between the labels participants produced for each new scene during the generalization test, and the labels they produced for familiar scenes during the memory test. A high generalization score reflects the fact that, given an unfamiliar scene, participants produced a label which was as similar as possible to the labels they produced for familiar scenes with similar features (e.g., the same shape and/or the same direction). That is, their labels during the memory and the generalization test followed the same principles. A low generalization score reflects the fact that, given an unfamiliar scene, participants produced a label which was different from the labels they produced for familiar scenes with similar features. That is, the labels they produced for unfamiliar scenes did not resemble those they produced in the memory test. Quantifying participants' generalization behavior was not a trivial task: this measure was a first attempt to quantify the complex realm of generalizations in artificial languages, and as such should be interpreted with caution. For a detailed discussion of this measure see Raviv (2020, pp. 189–191).

For each participant, the generalization score is the normalized correlation between (a) the pair-wise semantic distances between each new scene and all familiar scenes, and (b) the pair-wise string distances between each new label produced in the generalization test and all labels produced for familiar scenes during the memory test. This correlation was normalized to account for the fact that high-structure languages offer more possibilities to generalize to begin with. The generalization score is calculated in the following way: For each new scene in the generalization test, we first calculated the semantic differences between that new scene and all familiar scenes using Hamming distances, in the same way as described above for structure score and for guessing similarity. Second, we calculated the string differences between the new label produced for this scene and the labels produced for familiar scenes during the memory test using normalized Levenshtein distances, in the same way as described above for structure score and for production similarity. We repeated this calculation for all new scenes and their corresponding labels. Then, these two sets of pair-wise

distances (i.e., string distances and meaning distances between new and familiar scenes/labels) were correlated using the Pearson product-moment correlation. Finally, this correlation was scaled using a procedure inspired by the min-max normalization procedure (also called unity-based normalization and feature-scaling), yielding the final generalization score per participant. This normalization procedure was implemented in order to ensure that all conditions show similar ranges of generalization scores, and that we do not bias against low structured languages, which by default would show lower generalization scores given that participants' productions for familiar items are likely to be less structured in such languages. Specifically, we linearly transformed the correlation scores to fit in the range [0,1], and scaled across different conditions so that the final generalization score was proportionate to the range of achieved values in that condition: low generalization scores relative to the range of possible scores are mapped to values closer to 0, and high generalization scores relative to the range of achieved scores are mapped to values closer to 1. This was done using the formula $x' = (x-min(x))/(max(x)-min(x))$, where $min(x)$ in the lowest value for x achieved by a participant across all conditions ($-0.069$), and $max(x)$ is the highest value for x achieved by a participant in a specific condition (i.e., $max(x)$ varied for different input languages, with each input language having a different maximal value). For example, the highest value achieved by a participant in a low-structure language was 0.5, while the highest value achieved by a participant in a high-structure language was 0.88.

### 4.5. Generalization convergence

This continuous measure reflects the degree of similarity between the labels produced during the generalization test by different participants who learned the same input language. For each of the new scenes in the ten input languages, we calculated the normalized Levenshtein distances between all pairs of labels produced by different participants for the same new scenes. The average distance between all pairs of labels was subtracted from 1 to represent string similarity, i.e., how much the labels of different participants resembled each other. A high convergence score indicates that participants who learned the same language also produced similar labels for the unfamiliar scenes during the generalization test. A low convergence score indicates that participants who learned the same language produced different labels for unfamiliar scenes during the generalization test.

## 5. Analyses and Results

We analyzed the data using mixed effects regression models generated by the lme4 package in R (Bates et al., 2016; R Core Team, 2016). All reported p-values were generated using the pbkrtest package (Halekoh and Højsgaard, 2014), which uses the Kenward-Roger Approximation to calculate conservative p-values for models based a relatively small number of observations. All analyses are reported in Appendix C using numbered models, which we refer to here. The data and the R code to generate all analyses can be openly accessed at https://osf.io/d5ty7/.

### 5.1. Confirmatory analysis: Final Binary Accuracy (Fig. 2a)

As declared in the preregistration (under "Analysis Plan"), our main model had final binary accuracy (i.e., whether participants were right or wrong in the memory test) as the dependent variable, and included fixed effects for GROUP SIZE ORIGIN (dummy-coded, with small groups as reference level) and STRUCTURE SCORE (continuous, centered), as well as random intercepts for participants and scenes. Since we suspected that the effect of STRUCTURE SCORE would be non-linear (Beckner et al., 2017; Raviv et al., 2019b), we used Likelihood ratio tests to compare models with 1- and 2-degree polynomials (generated using the poly() function in R to avoid collinearity). These model comparisons revealed that the best fitting

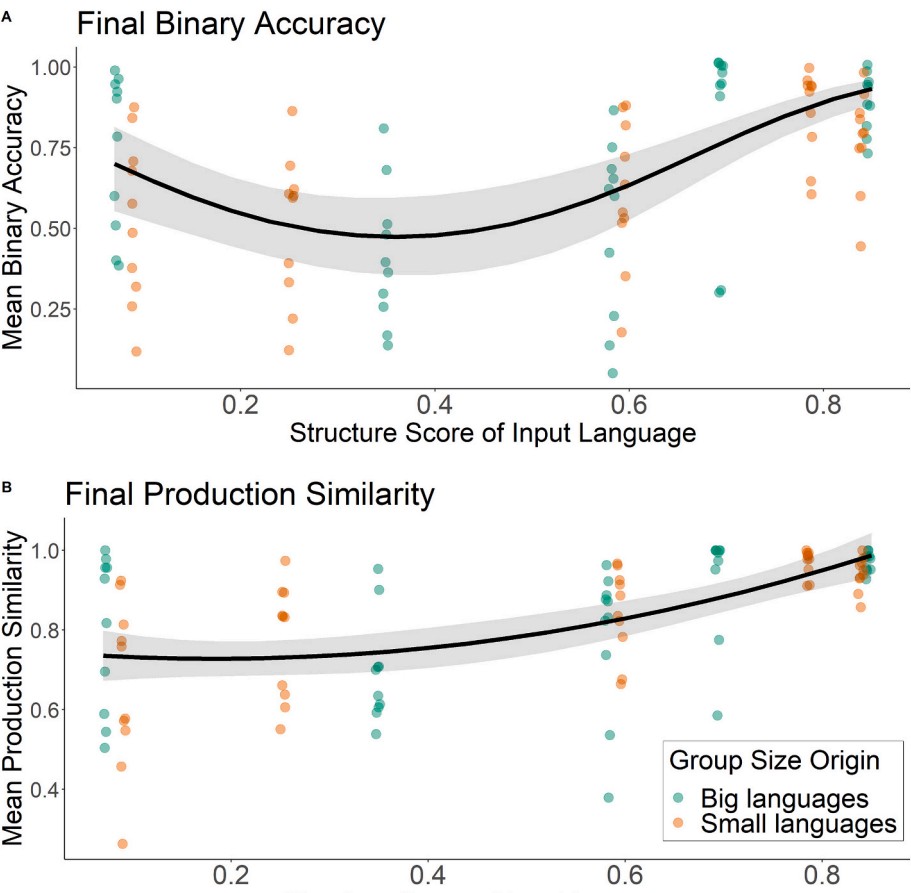

**Fig. 2.** (A) Binary Accuracy and (B) Production Similarity at the final memory test, as a function of learned languages' structure score and group size origin. Each point represents the average accuracy of a single participant. The thick line represent the model's estimate, and its shadings represent the model's standard error.

model (Model 1) included both a linear and a quadratic term (see Appendix C).

Results from this model showed that STRUCTURE SCORE was a positive significant predictor of participants' binary accuracy during the memory test (Model 1: β = 31.47, SE = 6.93, z = 4.54, p = 0.00001), and that this effect was non-linear (Model 1: β = 31, SE = 6.87, z = 4.51, p = 0.00001). Specifically, the effect of STRUCTURE SCORE on accuracy followed a U-shape: participants' binary accuracy was poorer when trained on medium structure languages than when trained on low structured languages, but the highest when trained on high structured languages (Fig. 2a). The U-shape pattern is evident in the global minimum of the polynomial function predicted by the model, which can be directly calculated when running the same model without the orthogonal polynomials and comparing its derivative to 0. After re-centering, we found that the minimum value for binary accuracy was obtained when structure equals 0.36, which is within the medium structure bin. In other words, participants' performance was worst when learning semi-structured languages, and the increase in structure only benefited accuracy as languages became highly systematic. The effect of GROUP SIZE ORIGIN was not significant, with languages originating from big and small groups eliciting similar levels of accuracy (Model 1: β = 0.48, SE = 0.29, z = 1.67, p = 0.096).

### 5.2. Exploratory analysis: Final Production Similarity (Fig. 2b)

Originally, we believed that binary accuracy was a good measure to examine learning, considering an "all or nothing" approach. However, during data collection we observed that this measure was too crude, and did not reliably reflect how well participants learned the languages.

Specifically, many participants were able to reproduce the language with relatively high fidelity - but not perfectly - which the binary accuracy measure did not capture. For example, if a participant correctly typed five letters out of a six-letter label, the binary accuracy measure would treat this one-letter error as if the entire label was incorrect. This led to an overestimation of errors, with some participants receiving low scores despite making very minor mistakes (e.g., one letter difference between the label they learned and the label they reproduced). As such, we decided to use a more subtle measure of participants' learning accuracy, namely, production similarity (see Measures). This continuous measure reflects the degree of reproduction accuracy more reliably by quantifying the similarity between participants' input and output languages, and is broadly used in iterated language learning paradigms (Kirby, Cornish, & Smith, 2008; Kirby, Tamariz, Cornish, & Smith, 2015). We therefore ran an identical model to that described in the confirmatory analysis section, but used production similarity during test as the dependent variable instead of binary accuracy during test. Importantly, the predictions for this measure were identical to those of binary accuracy: more structured languages should be reproduced more accurately, i.e., show more production similarity. Accordingly, the model for production similarity had the same effect structure as the binary accuracy model reported above, and included fixed effects for GROUP SIZE ORIGIN (dummy-coded, with small groups as reference level) and STRUCTURE SCORE (continuous, centered), and random intercepts for participants and scenes. As in the confirmatory analysis, Likelihood ratio tests favored the 2-degree polynomial model (Model 2) with a linear and a quadratic term for the effect of STRUCTURE SCORE (see Appendix C).

Results from this model showed that STRUCTURE SCORE was a positive significant predictor of production similarity during the memory test

(Model 2: β = 4.41, SE = 0.68, t = 6.49, p < 0.0001). This effect was also non-linear (Model 2: β = 1.6, SE = 0.68, t = 2.34, p = 0.02), yet in an exponential way: participants produced labels that were increasingly more similar to those they learned as structure increased, so that the advantage of structure was stronger in highly structured languages (Fig. 2b). That is, the increase in structure benefited accuracy most as languages became more systematic. As for binary accuracy, we calculated the global minimum of the polynomial function predicted by the model for production similarity, and found that the minimum value for reproduction fidelity was obtained when structure equaled 0.18, which is within the low structure bin. That is, participants' performance was worst when learning unstructured languages. The effect of GROUP SIZE ORIGIN was not significant, with languages originating from big and small groups eliciting similar levels of production accuracy (Model 2: β = 0.007, SE = 0.03, t = 0.26, p = 0.8).

### 5.3. Exploratory analyses: Learning Trajectory (Fig. 3)

As declared in the preregistration (under "Analysis Plan"), we also planned to perform an exploratory analysis to examine participants' learning trajectory during the three blocks of the learning phase. Specifically, we were interested in seeing whether improvement in performance during the first three blocks was modulated by structure score

and/or group size (e.g., are highly structured languages learned faster?). To this end, we generated three models in which the dependent variable was either binary accuracy, production similarity, or guessing similarity (see Measures). All three models had the same effects structure, and included fixed effects for BLOCK NUMBER (continuous, centered), GROUP SIZE ORIGIN (dummy-coded, with small groups as reference level), STRUCTURE SCORE (continuous, centered), and the interaction terms BLOCK NUMBER X GROUP SIZE ORIGIN and BLOCK NUMBER X STRUCTURE SCORE. All models included by-participant and by-scene random intercepts, as well as random by-participant slopes with respect to the effect of BLOCK NUMBER. We used Likelihood ratio tests to compare 1- and 2-degree polynomial models with respect to the effect of STRUCTURE SCORE (see Appendix C), and found that models with a quadratic term were favored in the case of binary accuracy (Model 3) and guessing similarity (Model 5), but not for production similarity (Model 4).

All three models yielded similar results (Fig. 3), and showed that performance significantly improved over learning blocks, with participants showing higher binary accuracy (Model 3: β = 0.29, SE = 0.05, z = 5.99, p < 0.0001), higher production similarity (Model 4: β = 0.04, SE = 0.007, t = 5.57, p < 0.0001) and higher guessing similarity (Model 5: β = 0.04, SE = 0.01, t = 3.74, p = 0.0003) over blocks. There was also a significant effect of STRUCTURE SCORE for all measures, indicating that across blocks, performance was overall better on more structured

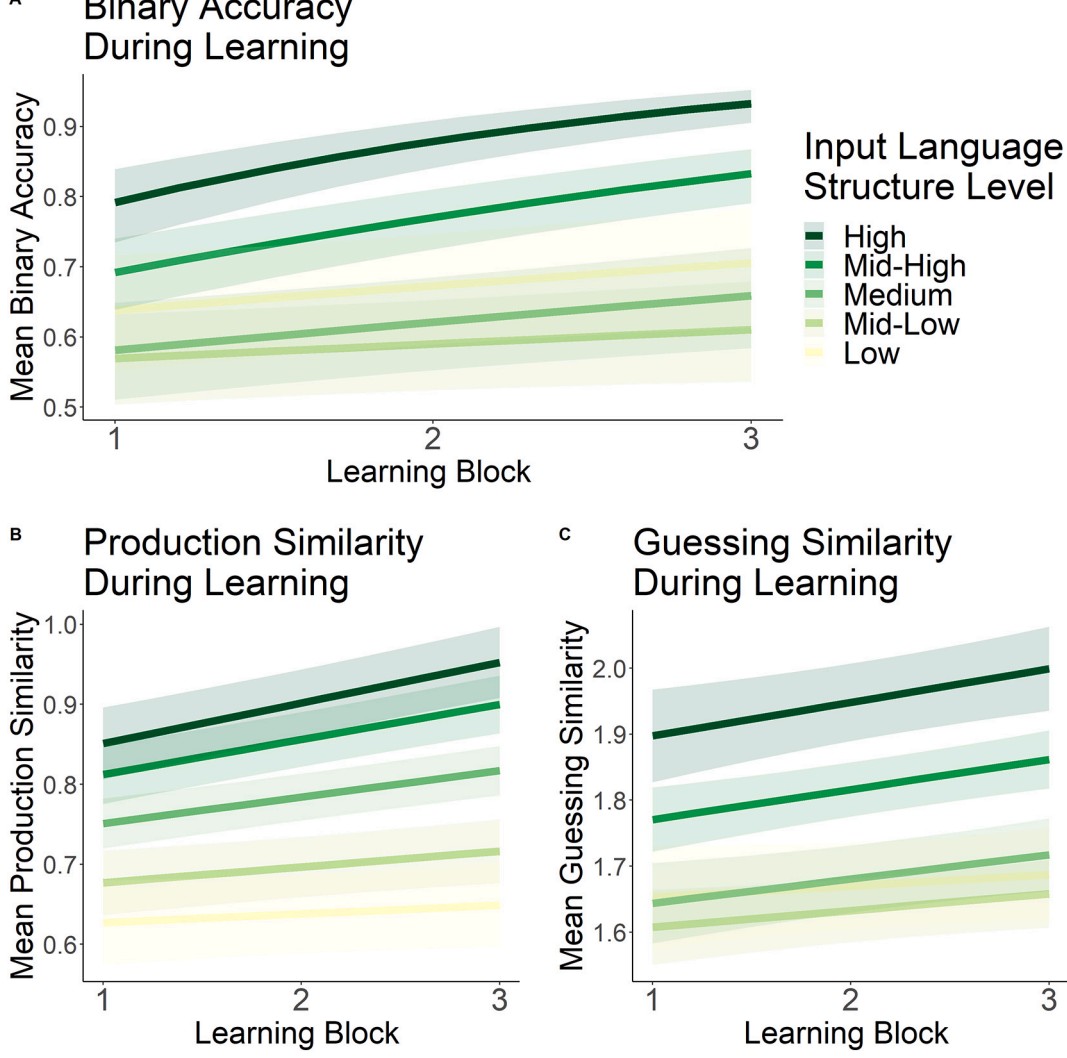

**Fig. 3.** Changes in Mean (A) Binary Accuracy, (B) Production Similarity, and (C) Guessing Similarity over learning blocks as a function of learned languages' structure score. The colored lines and their shadings represent the models' estimates and standard errors, averaged over the five descriptive structure levels (i.e., collapsed over big and small groups' languages).

languages (Model 3: β = 66.49, SE = 10.67, z = 6.23, p < 0.0001; Model 4: β = 0.34, SE = 0.05, t = 7.35, p < 0.0001; Model 5: β = 14.65, SE = 2.13, t = 6.88, p < 0.0001). This effect was non-linear for binary accuracy and guessing similarity, suggesting that the advantage of structure for these two measures was increasingly higher as structure increased (Model 3: β = 51.83, SE = 10.74, z = 4.83, p < 0.0001; Model 5: β = 8.07, SE = 2.14, t = 3.76, p = 0.0003). Additionally, there was a significant interaction between STRUCTURE SCORE and BLOCK NUMBER for binary accuracy (Model 3: β = 25.88, SE = 4.32, z = 6, p < 0.0001; β = 14.46, SE = 4.76, z = 3.04, p = 0.0024) and production similarity (Model 4: β = 0.05, SE = 0.02, t = 2.83, p = 0.00564), indicating that the improvement in participants' performance over blocks in these two measures was even faster for more structured languages, i.e., the learning slope was steeper for highly systematic languages. This interaction was not significant for guessing similarity (Model 5: β = 1.75, SE = 1.19, t = 1.48, p = 0.14), suggesting that the slope of improvement in participants' guessing performance over blocks was similar across all structural levels. Finally, GROUP SIZE ORIGIN did not significantly affect performance on any of our three measures (Model 3: β = 0.23, SE = 0.19, z = 1.19, p = 0.23; Model 4: β = 0.001, SE = 0.03, t = 0.04, p = 0.97; Model 5: β = 0.01, SE = 0.03, t = 0.41, p = 0.69) or on participants' learning trajectories (Model 3: β = 0.04, SE = 0.07, z = 0.54, p = 0.59; Model 4: β = −0.01, SE = 0.01, t = −1.37, p = 0.17; Model 5: β = −0.01, SE = 0.02, t = −0.86, p = 0.39).

### 5.4. Exploratory analyses: Generalization Behavior (Fig. 4)

As declared in the preregistration (under "Analysis Plan"), we also planned to examine participants' behavior during the generalization phase. In particular, we wanted to see whether participants would generalize the linguistic patterns of their input language to new, unseen scenes. If participants learned a systematic language and learned its underlying structure, generalizations could potentially take place in the form of reusing the learned structural patterns (i.e., part-words) when producing new labels (e.g., combining existing morphemes for shape and motion to describe a new scene with a new combination of shape and motion). If participants learned an unstructured language, generalizations could potentially take place in the form of reusing existing full words to describe scenes with similar elements (i.e., creating homonyms), or combining exiting words. In both cases, if the participants generalized their input language and maintained its patterns, then their productions for each new scene during the generalization test should be similar to their productions of the input language during the memory test, resulting in a high generalization score (see Measures). If participants did not generalize and instead produced random, unrelated labels, then their generalization score should be lower. This score was also adjusted to take into account the fact that low-structured languages allow for less generalizations to begin with. To test participants' generalization behavior, we used a general linear regression model with normalized generalization score as the dependent variable, and fixed effects for GROUP SIZE ORIGIN (dummy-coded, with small groups as reference level) and STRUCTURE SCORE (continuous, centered). We used Likelihood ratio tests to compare 1- and 2-degree polynomial models with respect to the effect of STRUCTURE SCORE, and found that the model with only a linear term (Model 6) was favored (Appendix C).

Results from this model showed that STRUCTURE SCORE was a significant predictor of generalization score: participants who had acquired more structured languages also generalized more (Model 6: β = 0.51, SE = 0.07, t = 7.22, p < 0.00001; Fig. 4). There was no significant effect of GROUP SIZE ORIGIN (Model 6: β = 0.01, SE = 0.04, t = 0.31, p = 0.76), suggesting that generalization behavior was similar for languages originating from big and small groups.

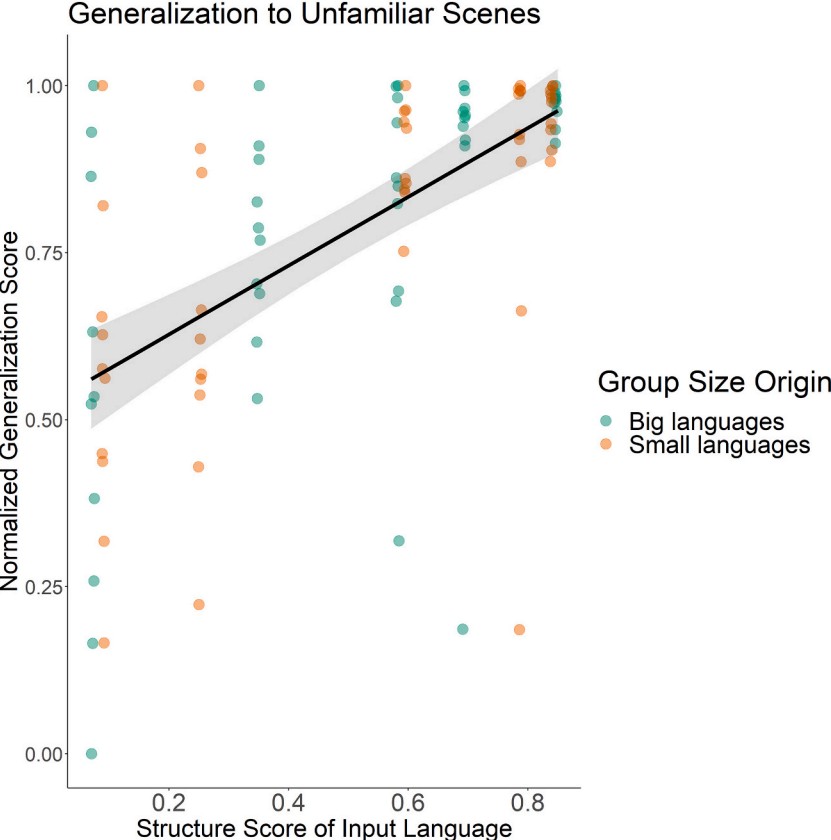

**Fig. 4.** Generalization as function of learned languages' structure score and group size origin. Each point represents the normalized generalization score of a single participant. The thick line represent the model's estimate, and its shadings represent the model's standard error.

### 5.5. Exploratory analyses: Generalization Convergence ([Fig. 5](#))

Finally, we looked for similarities in participants' generalizations: do participants in the same condition make similar generalizations, i.e., produce similar labels for unseen scenes? We assumed that when languages are highly systematic and rule-governed, they allow for transparent and productive labeling – resulting in different participants producing the same labels, i.e., generalizing in the same way. By contrast, when languages are unstructured or inconsistent in their mapping of labels to meanings, it may be less clear what or how to generalize (e.g., which features of the scenes are relevant?) and therefore less clear how to label new scenes. This may result in participants producing new labels more randomly, or attempting to make generalizations based on the idiosyncratic features of scenes (i.e., fill-pattern). In other words, we predicted that highly structured languages would facilitate convergence amongst participants, potentially enabling them to understand each other even without previously interacting. To test this prediction, we generated a mixed effect model with generalization similarity (convergence) on each unfamiliar scene as the dependent variable. The model had fixed effects for GROUP SIZE ORIGIN (dummy-coded, with small groups as reference level) and STRUCTURE SCORE (continuous, centered), and random intercepts with respect to scenes. We used Likelihood ratio tests to compare 1- and 2-degree polynomial models with respect to the effect of STRUCTURE SCORE, and found that the model with only a linear term (Model 7) was favored (see Appendix C).

Results from this model showed that STRUCTURE SCORE was a significant predictor of generalization score, so that participants who learned more structured languages also produced labels that were more similar to one another (Model 7: $\beta = 0.74$, SE $= 0.03$, $t = 21.63$, p $< 0.00001$; [Fig. 5](#)). There was no significant effect of GROUP SIZE ORIGIN (Model 7: $\beta = -0.03$,

SE $= 0.02$, $t = -1.71$, p $= 0.09$), suggesting that languages originating from big and small groups did not differ in their convergence.

### 6. Discussion

In this pre-registered study, we used an artificial language learning paradigm to test the effects of systematic structure and community size on language learnability by adults. We compared participants' acquisition of a broad yet controlled set of input languages for describing novel dynamic events (see [Fig. 1](#)). These input languages varied in their degree of linguistic structure (ranging from low to high systematicity) and in their group size origin (created by either big or small groups in a previous communication experiment, [Raviv et al., 2019b](#)). Language learnability was assessed by examining participants' final reproduction accuracy, their learning trajectories over blocks of training, and their ability to generalize the language they learned to a new set of unseen events.

Our main prediction was that participants would show better learning of languages with more systematic structures. This prediction was motivated by the literature reviewed in the Introduction (e.g., second language learning, iterated learning), which argued for a causal link between the grammatical structure of languages and their relative ease of learning, at least by adults. Specifically, more regular and transparent languages with more systematic form-to-meaning mappings are considered to be easier to learn. Therefore, we hypothesized that linguistic structure would positively affect learnability, such that languages with more systematic grammars would be better learned. We expected that this learning advantage would be reflected in higher accuracy during a memory test, and potentially also in a faster improvement in performance during the learning phase. Additionally, we

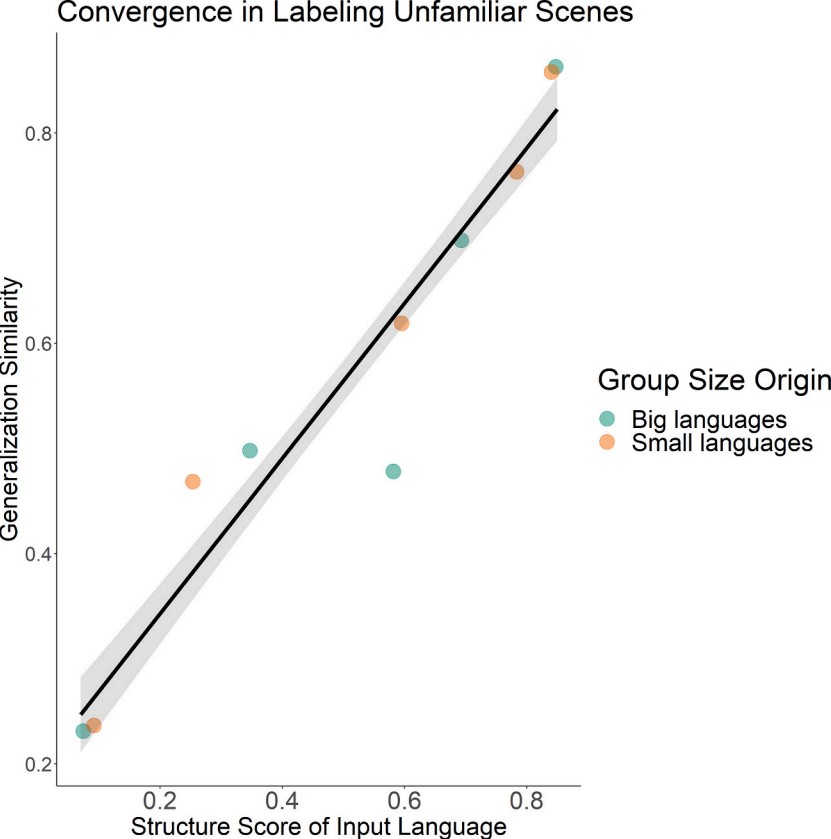

**Fig. 5.** Generalization convergence across participants as function of learned languages' structure score and group size origin. Each point represents the average convergence (i.e., label similarlity) between ten participants on each of the ten input languages. The thick line represent the model's estimate, and its shadings represent the model's standard error.

reasoned that systematic and rule-governed languages would allow for clear and productive labeling (Ackerman & Malouf, 2013; Kirby, 2002). Therefore, we predicted that more structured languages would be more easily generalizable to new meanings. To test this prediction, we assessed participants' ability to generalize the language they learned in order to produce new labels for unseen events. All of these predictions were borne out: adults were better and faster at learning languages with highly systematic structure, and produced more generalizations when learning such languages. We discuss these results in details below.

In addition, we asked whether there was an effect of group size beyond linguistic structure, such that languages that were created by bigger groups would be easier to learn even when equating for the degree of systematicity. We found no evidence that this was the case: across all measures and all analyses, we found no significant differences between the languages created by big and small groups. Notably, we cannot draw any strong conclusions from this null result. On one hand, it is possible that once the level of linguistic structure is controlled for, there are no additional benefits to learning languages created by big groups. In other words, the most relevant difference between big and small communities may, in fact, be their tendency to develop different degrees of systematicity (Raviv et al., 2019b; Lupyan and Dale, 2010). On the other hand, it is also possible that community size does affect language learnability beyond linguistic structure, but that we did not capture this difference in the current study. Specifically, the lack of significant group size effects in our study may be attributable to our stimuli selection procedure: when selecting the input languages for this experiment, we intentionally controlled for several linguistic features that may make languages more or less learnable above and beyond their structure score (i.e., word length, confusability, number of irregulars). It is possible that these selection criteria incidentally washed away relevant differences between the two group size conditions that could affect the languages' learnability. For example, perhaps word length was, in fact, one of the features that differentiate the languages created by big and small groups, and by controlling for it we have eliminated relevant variation. We discuss these possibilities in detail in Raviv (2020, pp. 192–195). In sum, it is currently unknown whether group size impacts learning beyond the effect of linguistic structure, and more research is needed in order to confirm or refute this possibility.

With respect to our main prediction, the results from the confirmatory analysis showed that the relationship between language learnability and linguistic structure followed a U-shape (Fig. 2a): although participants' mean accuracy was, as predicted, highest when learning highly structured languages, it was poorest when learning medium structured languages, and not when learning low structured languages (as one would expect if the relationship between structure and learning was simply linear). That is, learners struggled most with learning languages that were partly or semi-structured, i.e., languages that contained some patterns but also multiple irregulars and inconsistencies. This pattern, however, was not fully replicated in a similar exploratory model, where we examined participants' learning by using a more subtle measure of reproduction fidelity (i.e., production similarity) that reflected the degree of similarity between the labels participants learned and the labels they eventually reproduced. Results from this model also supported a non-linear relationship between structure and learnability, albeit an exponential relation and not U-shaped: participants produced more similar labels to those they learned as linguistic structure increased, and especially so for highly compositional languages (Fig. 2b). In other words, the benefit of linguistic structure for learning was proportionate to the level of structure in the language, and increased as structure increased. Similar findings were obtained from a set of exploratory analyses that investigated participants' learning trajectories over the course of the experiment: participants' performance was better on more structured languages throughout the training phase, and gradually improved across learning blocks. Moreover, the reproduction accuracy of participants who learned highly structured languages improved more quickly.

Together, our results confirm that a higher degree of linguistic structure is advantageous for language learning, and show that languages with highly structured grammars are learned faster and more accurately by adults. These findings are in line with our main prediction, and corroborate the postulated link between the degree of systematicity in the language and its relative learnability by adult learners. This link is important for theories of language evolution and language diversity discussed in the Introduction, which rely on it as an explanatory mechanism. Although the non-linear nature of the relationship between language structure and language learnability warrants further explanation (which we discuss below), our results do support a causal relationship between them: highly regular and systematic morphologies indeed seem easier to learn for adults. This conclusion strengthens the premise that not all grammatical systems are equally learnable (at least not by adults in a limited time), and has broader implications for theories on second language learning. Specifically, our study supports the claim that cross-linguistic differences in structural complexity and morphological opacity can result in different proficiency levels for adult L2 learners learning different languages.

In our view, these results can potentially extend to language acquisition by children. This would imply that children's language learning trajectories may also differ cross-linguistically depending on the degree of systematicity in their native language (Slobin, 1985). Even though children and adults differ substantially in their learning biases, learning strategies, and generalization patterns (Culbertson and Newport, 2015; Hudson Kam & Newport, 2005, 2009; Newport, 2020; Schuler, 2017), we predict that such differences would not cancel out the positive effect of systematicity on learning and memorization. Because the positive link between systematicity and learnability is based on general principles of compressibility and simplicity (e.g., Chater and Vitányi, 2003; Culbertson and Kirby, 2016; Kortman, 1967), it is highly likely that children would also benefit from the presence of more regularity in their input. The possibility that children would *not* benefit from regularity at all (i.e., showing a flat learning curve across all levels of structure), or that they would show the opposite pattern altogether (i.e., better learning of low-structured languages) seems unlikely. If anything, the fact that children have a much stronger bias in favor of regularity and tend to generalize more than adults (e.g., Hudson Kam & Newport, 2005) suggests that children may benefit even *more* from systematic linguistic structure. Future work should test this hypothesis directly by examining whether more structured languages are also learned better by children.

Our results also show that systematic structure can be advantageous for making generalizations: in an exploratory analysis, we found that participants generalized significantly more as linguistic structure in their input language increased. Specifically, participants who learned more systematic languages created new labels that matched the patterns of their input language more closely. This finding suggests that, in addition to being beneficial for learning and memorization, an important advantage of linguistic structure is its productivity. That is, learners can potentially exploit transparent, systematic, and regular patterns found in their language to make informed guesses about unknown forms of words based on exposure to known forms, allowing them to effectively produce new labels for unfamiliar meanings. However, given that these results were based on a preliminary, exploratory measure, they should be taken with caution and require further experimental validation. Specifically, there was no prior measure of generalizations in artificial languages that we could rely on, and as such it was not clear how best to quantify it (especially in low structured languages with no obvious structure). Moreover, we cannot be sure what the overall distribution of possible generalization scores was, and whether it was similar across different input language conditions. We discuss the potential limitations of this measure in (Raviv, 2020, pp. 189–191). Nevertheless, the advantage of systematicity for generalizations was also evident when looking at participants' self-reported behavior in the final questionnaire: all participants learning languages with high systematic structure indicated that they "knew" how to label the new scenes in the

generalization test, and some of them did not even notice that these scenes did not appear during training.

In addition to being beneficial for individuals' generalization behavior, high structure languages were advantageous for potential communication between individuals, i.e., convergence. When we examined the new labels produced for unseen events by different participants who learned the same language, we found that participants who learned more structured languages produced labels that were significantly more similar. That is, systematic structure led different participants to produce similar labels for new meanings without previously learning them and without previously interacting with each other. This finding suggests that systematicity can allow strangers to converge effortlessly: people who never interacted before could potentially communicate successfully about new events – and immediately be understood. This finding supports the postulated mechanism behind larger communities' tendency to develop more systematic languages (Meir, Israel, Sandler, Padden, & Aronoff, 2012; Meir and Sandler, 2019; Raviv et al., 2019b). Small communities typically have tightly connected networks of individuals who are highly familiar with each other (Granovetter, 1983), and can therefore rely on common ground and shared history when communicating about novel events. In contrast, bigger communities have more strangers (i.e., individuals rarely or never interact; Granovetter, 1983), who cannot rely on shared history to support mutual understanding. Nevertheless, they need to be able to understand each other when interacting for the first time. As such, it was argued that members of bigger communities are under a stronger pressure to develop transparent, predictable, and systematic structures that aid convergence and allow strangers to successfully communicate (Nettle, 2012; Wray and Grace, 2007; Raviv et al., 2019b). Although we cannot draw any direct causal inferences from the current study, our findings support this hypothesis by showing that the benefits of systematic linguistic structure go beyond learnability, and that systematicity can aid communication and productivity in general language use.

Notably, the relationship between linguistic structure and language learnability was not a straight-forward, linear relationship. Although we did predict that this relationship may be non-linear (e.g., that it would be stronger or weaker as structure increases), we were not expecting a U-shape pattern where completely unstructured languages are easier to learn than medium structured languages. Rather, we hypothesized that holistic languages with no systematic structure whatsoever would be harder to learn than languages that exhibit *some* systematic structure, i. e., that any increase in structure would be advantageous for learning. Counterintuitively, participants' final binary accuracy suggested that the hardest languages to learn were those that exhibit some structure, as opposed to none. Even when looking only at participants' final production fidelity, it was not the case that completely holistic and unstructured systems were harder to learn. Rather, low-structured languages and medium-structured languages showed similar production fidelity. However, this nonlinear pattern may not actually hold in natural languages, and might not faithfully represent speakers' true learning biases. Real-world natural languages are never truly holistic or structure-free: there are no known languages which are fully suppletive or consist only of unpredictable inflections (Ackerman & Malouf, 2013). Instead, natural languages are inherently quasi-regular, and typically consist of some regular and transparent patterns alongside pockets of opacity and exceptions to the rule (Kempe & Brooks, 2008). Since the low-structure languages in our experiment do not really resemble natural languages, it is possible that the non-linear relationship we observed between language learnability and linguistic structure was caused by our artificial choice of stimuli, and does not extend to the real-world. If natural languages realistically range only from medium-structure to high-structure, then the actual relevant relationship between systematicity and learning may indeed be linear.

While it is possible that a non-linear relationship between language learnability and grammatical structure is less relevant for natural language environments as explained above, the nonlinear result obtained in this study (i.e., that partly structured languages are not easier to learn than unstructured languages) is still puzzling. In particular, our original expectation was based on findings from two artificial language learning studies that examined the benefit of systematic sound-mapping for learning (Brooks et al., 1993; Monaghan et al., 2011). In those studies, languages with partially consistent mapping between phonological features and noun classes were learned better than completely arbitrary languages. Importantly, the stimuli used in those studies can also be seen as unrepresentative of natural languages, given that all natural languages have some degree of iconicity and are never completely arbitrary (Perlman, Little, Thompson, & Thompson, 2018). Yet despite the equally artificial nature of their stimuli, those studies showed that partial systematicity did aid learning. As such, the unnaturalness of fully unstructured languages does not exempt us from explaining this unpredicted pattern and the discrepancy from previous studies.

A tentative explanation for the nonlinear relationship we found between learnability and systematicity is that, although partial structure can provide some regularity in the form of statistical cues for meaning, it might also result in more uncertainty and a high cognitive load for learners. The inconsistent patterns in medium structured languages may be similarly or even more confusing to learn than a set of unrelated words given (a) participants' learning strategies, and (b) cue validity. First, let us consider that learners are trying (explicitly or implicitly) to build hypotheses about potential linguistic rules (MacWhinney, 1978). This idea is supported by studies showing that speakers automatically attempt to decompose pseudo-words and non-words into smaller components in a lexical similarity task (Post et al., 2008): any stimulus that can be potentially interpreted as ending in an inflection, whether real or not, is responded to more slowly than an unambiguous stimulus. Moreover, adults tend to assume that unpredictable variation is, in fact, meaningful, and tend to treat random patterns as if they depend on factors not yet discovered (Perfors, 2016). Such findings suggest that speakers try to figure out the underlying structure of word forms, and that morphosyntactic ambiguity can therefore elicit processing costs and learning difficulties when these hypotheses are not met Consequently, participants' learning strategy may differ across conditions. Let us assume that all learners start out with an item-based learning strategy and initially rely on memorizing individual words. Learners exposed to highly systematic languages may gradually detect consistent patterns in their input (i.e., part-words that are reguarly associated with semantic features) and consequently form rules and abstractions (Kempe & Brooks, 2008; Tomasello, 2000). However, this switch from item-based learning to rule-learning can only occur when there *are* productive rules in the language, i.e., when forming a generalization is more efficient than storing lexical forms individually. In low-structured languages, there are no such productive rules: there are no meaningful or useful patterns in the language, and word forms appear at random. After several attempts to form abstractions, learners of completely unstructured languages may realize that there are always more irregulars than regulars no matter which rule they form, and therefore abandon the search altogether. In this case, learners of low structured languages may, by hypothesis, "give up" on looking for rules after a short period of time, and simply focus on memorizing the holistic lexicon in a purely item-based manner (i.e., rote learning). This process is evident in natural languages too, where some grammatical features are random and require learners to simply memorize them (e.g., the grammatical gender of names for inanimate objects in German). But crucially, learners of medium structured languages may face an especially challenging task since they need to realize that only some, but not all, of their input is systematic (e.g., shapes may have consistent markings, but angles don't). Learners of such languages may be motivated to keep looking for systematic cues and abstractions, even when these do not exist. The fact that their input contains regularities with respect to some features (i.e., shape) but not others may lead to more confusion and even frustration when compared to learners of low structured languages. Note that even if we abandon such a rule-based learning model in favor of associative

learning, i.e., learning as gradual strengthening of the association between co-occurring elements of the language, the absence of valid and reliable cues would still hinder learning (Kempe & MacWhinney, 1998).

In any case, our finding that the relation between linguistic structure and learnability was not linear in adult learners (i.e., so that holistic languages are not necessarily more difficult to learn for adults) poses a potential problem for iterated language learning models, which rely on a learning advantage of *some* structure compared to none (Kirby, Cornish, & Smith, 2008; Kirby, Tamariz, Cornish, & Smith, 2015; Smith, 2011). Specifically, studies on the cultural evolution of compositionality via iterated learning have shown that compositional linguistic structure gradually emerges from a state of a holistic lexicon over multiple generations of learners, be it simulated computer agents, adult participants, or children. Crucially, this slow accumulation in structure is typically attributed to learnability pressures, i.e., to agents' difficulty in memorizing a completely unstructured lexicon. Accordingly, such models predict that the learning advantage provided by linguistic structure should already be present in the very early stages of language evolution, such that even the smallest degree of added regularity in the input should facilitate learning. One way to reconcile these claims with our findings is to argue that creating linguistic structure has additional benefits to language users, above and beyond the benefits to memory. Indeed, our study suggests that this is the case: highly systematic languages are favored not only because they are more learnable by adults, but also because they are more predictable and allow for clearer generalizations and quick convergence between individuals. This idea resonates with early iterated learning models (Kirby, 2002; Smith et al., 2003), which stress the benefit of linguistic structure for generalizations: although agents are usually not exposed to the entire repertoire of the language, they must be able to produce labels to new events despite their partial exposure. Another possibility is that the relationship between learnability and systematicity is only non-linear in adults but not in children: while adult learners do not seem to benefit from the existence of partial regularities in their input, children might. This hypothesis is in line with studies showing a stronger bias towards regularization and generalizations in children compared to adults (e.g., Hudson Kam & Newport, 2005): children might be even more sensitive to systematicity than adults and may generalize their input even more.

## 7. Conclusions

The current study tested the acquisition of different artificial languages that varied in their degree of systematic structure and in their community size origin. We found that more linguistic structure generally benefited adults' language learning, with highly structured languages learned fastest and most accurately. At the same time, the relationship between language learnability and linguistic structure was not straight forward: high systematicity was indeed advantageous for learning, but adults did not seem to benefit from partly or semi-structured languages (i.e., languages that contained some patterns but also multiple irregulars and inconsistencies). Crucially, our results suggest that systematic structure is not only beneficial for memory and learning, but also for generalizations and convergence. Participants who learned more structured languages were better at generalizing the language they learned to new, unfamiliar meanings. Moreover, different participants tended to create similar new labels as structure increased. That is, more systematicity facilitated convergence and mutual understanding between strangers. Finally, we found no evidence that community size affected learnability beyond the degree of systematic structure: The languages that evolved in big and small groups were not significantly different in how accurately or how quickly they were learned.

## Author contribution

L.R. designed the research, analyzed the data, and wrote the paper;

M.D.H.K. designed and performed the research;
A.M. designed the research;

## Acknowledgments

We wish to thank Caitlin Decuyper for help with programming the experiment, Laurel Brehm for help with the power analysis, Dennis Joosen for help with data collection, and Ashley Micklos, Bill Thompson, Gary Lupyan,Jelle Zuidema, Tessa Verhoef, and Vera Kempe for their valuable input.

## Appendix A. Supplementary data

Supplementary data to this article can be found online at https://doi.org/10.1016/j.cognition.2021.104620.

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
