# OpenReview forum: "What makes a language easy to learn? A preregistered study on how systematic structure and community size affect language learnability"
_ICLR.cc/2022/Workshop/EmeCom — EmeCom Workshop at ICLR 2022_

### Official Review · Reviewer_QW7z · 2022-03-21
**Interesting paper with human study**

**Rating:** Accept
**Confidence:** 5

**Review:**

While I am not an expert in cognitive scientist, I am familiar with the topic on language learnability and its connection to ease-of-learning. This paper mainly provides a confirmation for this statement with the emerged language from another study. I think it is a good contribution for the workshop so that more people are aware of this statement on linguistic structure and ease-of-learning.

---

### Official Review · Reviewer_8Xbf · 2022-03-23

**Rating:** Accept
**Confidence:** 3

**Review:**

Summary of the contributions:

This paper investigates the impact of two variables on language acquisition, with human subjects. The two variables are the amount of structure (as measured by a metric akin to topographic similarity [Brighton and Kirby, 2006]) in the artificial languages whose participants have to consider, and the population size that made those artificial languages emerge (as found in a previous study).

The paper reports on the learnability/memorisation/ease-of-learning (measured via a memorisation test on the training set) and the generalisation abilities (measured via a zero-shot compositional test, as it is common in the language emergence field) of the participants depending on the artificial language they were considering.


Decision:

Given the goal of this year’s workshop to foster inter-disciplinary discussions, I think that this paper is very relevant as it tackled some concerns that have been investigated in the field of language emergence with populations of artificial agents and it does so with populations of human participants.

References:

H. Brighton and S. Kirby. Understanding Linguistic Evolution by Visualizing the Emergence of Topographic Mappings. Artificial Life, 12(2):229–242, jan 2006. ISSN 1064-5462. doi: 10.1162/artl.2006.12.2.229. URL http://www.mitpressjournals.org/doi/10.1162/artl.2006.12.2.229.

---

### Decision · Program_Chairs · 2022-03-25

**Decision:**

Accept

**Comment:**

This previously published work gives a great cognitive science/language evolution perspective on ease of language learning and we look forward to having discussions at the workshop!